

**Development and evaluation of a system of proxy data assimilation for**
**paleoclimate reconstruction**
By
Atsushi Okazaki[1] and Kei Yoshimura[2]
[1]RIKEN Advanced Institute for Computational Science, Japan
[2]Institute of Industrial Science, The University of Tokyo, Japan
*Submitted to Climate of the Past*
*Submitted in November, 2016*
*Revised in February, 2017*
______________________
Corresponding author: Atsushi Okazaki, RIKEN Advanced Institute for Computational
Science, 7-1-26 Minatojima-minami-machi, Chuo-ku, Kobe, Hyogo 650-0047, Japan
(atsushi.okazaki@riken.jp)

**Abstract**

Data assimilation (DA) has been successfully applied in the field of paleoclimatology
to reconstruct past climate. However, data reconstructed from proxies have been
assimilated, as opposed to the actual proxy values. This banned to fully utilize the
information recorded in the proxies.
This study examined the feasibility of proxy DA for paleoclimate reconstruction.
Isotopic proxies ($\delta^{18}$O in ice cores, corals, and tree-ring cellulose) were assimilated into
models: an isotope enabled general circulation model (GCM) and forward proxy models,
using offline data assimilation.
First, we examined the feasibility using an observation system simulation experiment
(OSSE). The analysis showed a significant improvement compared with the first guess in
the reproducibility of isotope ratios in the proxies, as well as the temperature and
precipitation fields, when only the isotopic information was assimilated. The
reconstruction skill for temperature and precipitation was especially high at low latitudes.
This is due to the fact that isotopic proxies are strongly influenced by temperature and/or
precipitation at low latitudes, which, in turn, are modulated by the El Niño-Southern
Oscillation (ENSO) on interannual timescales.
Subsequently, the proxy DA was conducted with real proxy data. The reconstruction
skill was decreased compared to the OSSE. In particular, the decrease was significant
over the Indian Ocean, eastern Pacific, and the Atlantic Ocean where the reproducibility
of the proxy model was lower. By changing the experimental design in a stepwise manner,
the decreased skill was suggested to be attributable to the misrepresentation of the
atmospheric and proxy models and/or the quality of the observations. Although there
remains a lot to improve proxy DA, the result adequately showed that proxy DA is
feasible enough to reconstruct past climate.

## 1.    Introduction

Knowledge of past conditions is crucial for understanding long-term climate variability. Historically, two approaches have been used to reconstruct paleoclimate; one based on the empirical evidence contained in proxy data, and the other based on simulation with physically-based climate models. Recently, an alternative approach combining proxy data and climate simulations using a data assimilation (DA) technique has emerged. DA has long been used for forecasting weather and is a well-established method. However, the DA algorithms used for weather forecasts cannot be directly applied to paleoclimate due to the different temporal resolution, spatial extent, and type of information contained within observation data (Widmann et al., 2010). The temporal resolution and spatial distribution of proxy data are significantly lower (seasonal at best) and sparser than the present-day observations used for weather forecasts, and the information we can get does not measure the direct states of climate (e.g., temperature, wind, pressure, etc.), but represents proxies of those states (e.g., tree-ring width, isotopic composition in ice sheets, etc.). Thus, DA applied to paleoclimate is only loosely linked to the methods used in the more mature field of weather forecasting, and it has been developed almost independently from them.

Several DA methods have been proposed for paleoclimate reconstruction (von Storch

et al., 2000; van der Schrier et al., 2005; Dirren and Hakim, 2005; Goosse et al., 2006;
Bhend et al., 2012; Dubinkina and Goosse, 2013; Steiger et al., 2014), and paleoclimate
studies using DA have successfully determined the mechanisms behind climate changes
(Crespin et al., 2009; Goosse et al., 2010; 2012; Mathiot et al., 2013). In previous studies,
the variables used for assimilation have been data reconstructed from proxies (e.g.,
surface air temperature) because observation operators or forward models for proxies
have not been readily available. Hereafter, the DA method that assimilates reconstructed
data from proxies is referred to as reconstructed DA. Recently, proxy modelers have
developed and evaluated several forward models (e.g., Dee et al., 2015 and references
therein). Thanks to that, currently a few studies have started attempting to assimilate
proxy data directly (Acevedo et al., 2016; Dee et al., 2016).
The main advantage of proxy DA over reconstructed DA is the richness of information
used for assimilation. In previous studies, only a single reconstructed field was
assimilated. However, proxies are influenced by multiple variables. Hence, the
assimilation of a single variable does not use the full information recorded in the proxies.
The reconstruction method itself also limits the amount of information. The most
commonly-used climate reconstruction is an empirical and statistical method that relies
on the relationships between climate variables and proxies observed in present-day
observations. These relationships are then applied to the past climate proxies to
reconstruct climate prior to the instrumental period. Most of the studies using this
approach assume that the relationship is linear. However, this assumption imposes
considerable limitations in which specific climate proxies can be used, and proxies that
do not satisfy the assumption have generally been omitted (e.g., PAGES 2k Consortium,
2013). Because information on paleoclimate is scarce, it is desirable to use as much
information as possible.
Furthermore, the reconstruction method also limits the quality of information
provided. The method also assumes stationarity of the relationship between the climate
and the proxies. However, this assumption has been shown to be invalid for some cases
(e.g., Schmidt et al. 2007; LeGrande and Schmidt, 2009). In the case of reconstructed DA,
the assimilation of such questionable reconstructed data would provide unrealistic results.
In the case of proxy DA; however, the skill of the assimilation is expected to be unchanged,
provided the model can correctly simulate the non-stationarity.
The concept of proxy data assimilation is not new, and has been proposed in previous
studies (Hughes and Ammann, 2009; Evans et al., 2013; Yoshimura et al., 2014; Dee et
al., 2015). Yoshimura et al. (2014) demonstrated that the assimilation of the stable water
isotope ratios of vapor improves the analysis for current weather forecasting. They
performed an observation system simulation experiment (OSSE) assuming that isotopic
observations from satellites were available every six hours. Because the isotope ratio of
water is one of the most frequently used climate proxies, this represents a significant first
step toward improving the performance of proxy data assimilation in terms of identifying
suitable variables for assimilation. However, it is not yet clear whether it is feasible to
constrain climate only using isotopic proxies whose temporal resolution and spatial
coverage are much longer and sparser than those of the specific study.
This study examined the feasibility of isotopic proxy DA for the paleoclimate
reconstruction on the interannual timescale. Because the study represents one of the first
attempts to assimilate isotopic variables on this timescale, we adopted the framework of
an OSSE, as in previous climate data assimilations (Annan and Hargreaves, 2012; Bhend
et al., 2012; Steiger et al., 2014; Acevedo et al., 2016b; Dee et al., 2016). After the
evaluation of proxy DA in the idealized way, we conducted the study with "real" proxy
DA. We investigated which factors decreased or increased the skill of the proxy DA. As
a measure of skill, we report the correlation coefficient throughout the manuscript.
In this study, we used only oxygen isotopes ($^{18}$O) as proxies. The isotope ratio is
expressed in delta notation ($\delta^{18}$O) relative to Vienna Standard Mean Ocean Water
(VSMOW) throughout the manuscript. If the original data were expressed in delta
notation relative to Vienna Pee Dee Belemnite (VPDB), they were converted to the
VSMOW scale.

This paper is structured as follows. In the following section, the data assimilation

algorithm, models, data, and experimental design are presented. Section 3 shows the
results of the idealized experiment. Section 4 gives the results of the real proxy DA. The
Discussion is presented in Section 5. Finally, we present our conclusions in Section 6.

**2.        Materials and methods**
**2.1.     Data assimilation algorithm**

We used a variant of ensemble Kalman filter (EnKF, see Houtekamer and Zhang, 2016,

and references therein); sequential ensemble square root filter (EnSRF; Whitaker and
Hamill, 2002). EnSRF updates the ensemble mean and the anomalies from the ensemble
mean separately, and processes observations serially one at a time if the observations have
independent errors.

To assimilate time-averaged data, slight modification was made for the method

following Bhend et al. (2012) and Steiger et al. (2014). In the modified EnSRF, the
analysis procedure is not cycled to the simulation (Bhend et al., 2012); thus, the
background ensembles can be constructed from existing climate model simulations
(Huntley and Hakim, 2010; Steiger et al., 2014). As such, we can assimilate data with any
temporal resolution coarser than the model outputs. In this study, we focused on annual
DA.

There are two ways to construct the background ensemble in the approach mentioned

above (hereafter offline DA); one using ensemble runs as in weather forecasts (Bhend et
al., 2012; Acevedo et al., 2016) and the other using a single run (Steiger et al., 2014; Dee
et al., 2016). The latter uses the same background ensemble for every analysis step. To
reduce computational cost, we chose the latter way, where the ensemble members are
individual years. This simplification was valid because the interannual variability in a
single run was inherently indistinguishable from the variability in the annual mean within
the ensemble of simulations in which the initial conditions were perturbed, at least for
atmospheric variables. Thus, the background ensembles were the same for all the
reconstruction years and did not contain any year-specific boundary conditions and
forcing information; hence, the background error covariance was constant over time.
Therefore, this study did not consider non-stationarity between the proxies and climate.
Despite the limitations of the algorithm used in this study, it should be noted that the
proxy DA could address non-stationarity if one uses temporally varying background
ensemble. We return to this point in Section 5.
To control spurious long-distance correlations due to sampling errors, a localization
function proposed by Gaspari and Cohn (1999) with a scale of 12,000 km was used. The
detailed procedure used for the algorithm is described in Steiger et al. (2014).


**2.2.    Models**
Isotope ratios recorded in ice cores, corals, and tree-ring cellulose were assimilated.
To assimilate these variables, forward models for the variables are required. We used the
forward model developed by Liu et al. (2013; 2014) for corals, and Roden et al. (2000)
for tree-ring cellulose. We assumed that the isotopic composition of ice cores was the
same as that of precipitation at the time of deposition. Note that, in reality, the isotope
ratio recorded in ice cores is not always equal to that in precipitation due to post-
depositional processes (e.g., Schotterer et al., 2004). Because detailed models that
explicitly simulate the impact of all the processes involved in determining the value of
the ratio are not yet available, we used the isotope ratio in precipitation for that in ice
cores to avoid adding unnecessary noise.
The isotopic composition in precipitation was simulated using an atmospheric general
circulation model (GCM) into which the isotopic composition of vapor, cloud water, and
cloud ice are incorporated as prognostic variables. The model explicitly simulates the
isotopic composition with all the details of the fractionation processes combined with
atmospheric dynamics and thermodynamics, and hydrological cycles. Hence, the model
simulates the isotopic composition consistent with the modeled climate. Although many
such models have been developed previously (Joussaume et al., 1984, Jouzel et al., 1987;
Hoffmann et al., 1998; Noone and Simmonds, 2002; Schmidt et al., 2005; Lee et al., 2007;
Yoshimura et al., 2008; Risi et al., 2010; Werner et al., 2011), we used a newly-developed
model (Okazaki et al., in prep.) based on the atmospheric component of MIROC5
(Watanabe et al. 2010). The spatial resolution was set to T42 (approximately 280 km)
with 40 vertical layers.

The variability in $\delta^{18}O$ recorded in coral skeleton aragonite ($\delta^{18}O_{coral}$) depends on the

calcification temperature and local $\delta^{18}O$ in sea water ($\delta^{18}O_{sw}$) at the time of growth
(Epstein and Mayeda, 1953). Previous studies have modeled $\delta^{18}O_{coral}$ as the linear
combination of sea surface temperature (SST) and $\delta^{18}O_{sw}$ (e.g., Julliet-Leclerc and
Schmidt, 2001; Brown et al., 2006; Thompson et al., 2011), as follows:

$$\delta^{18}O_{coral} = \delta^{18}O_{sw} + aSST \quad (1)$$

where $a$ is a constant which represents the slope between $\delta^{18}O_{coral}$ and SST. In this study,
the constant was uniformly set to -0.22‰/°C for all the corals, following Thompson et al.
(2011), and we used a model developed by Liu et al. (2013; 2014) to predict $\delta^{18}O_{sw}$. The
model is an isotopic mass balance model that considers evaporation, precipitation, and
mixing with deeper ocean water. The coral model uses the monthly output of the isotope-
enabled GCM as its input, except for the isotope ratio of deeper ocean water, which was
obtained from observation-based gridded data compiled by LeGrande and Schmidt et al.
(2006). After the model calculates the monthly $\delta^{18}O_{coral}$, it is arithmetically averaged to
provide the annual $\delta^{18}O_{coral}$.

The isotope ratio in tree-ring cellulose ($\delta^{18}O_{tree}$) was calculated using a model

developed by Roden et al. (2000). In this model, $\delta^{18}O_{tree}$ is determined by the isotopic
composition of the source water used by trees for photosynthesis, and evaporative
enrichment on leaves via transpiration. In this study, the value of the isotopic composition
in the source water was arbitrarily assumed to be the moving average, traced three-months
backward, of the isotopic composition in precipitation at the site. Again, the model used
the monthly output of the isotope-enabled GCM as its input. After performing the tree-
ring model calculation, the monthly output was weighted using climatological net primary
production (NPP) to calculate the annual average. The NPP data were obtained from the
US National Aeronautics and Space Administration (NASA) Earth Observation website
(http://neo.sci.gsfc.nasa.gov).
Because the isotopic compositions of the proxies were simulated using the output of
the isotope-enabled GCM, their horizontal resolution was the same as that of the GCM.

**2.3.     Experimental design**
**2.3.1.   Control experiment**
The first experiment served as a control (CTRL) experiment, and used the framework
of an OSSE. In the experiment, the "simulation" and the "truth" (nature run) were
simulated by the same models, with the same forcing, but with different initial conditions.
Because the proxy models were driven by the output of the GCM, the modeled proxies
were consistent with the modeled climate from the GCM. Thus, here we describe the
experimental design for the GCM. The GCM was driven by observed SST and sea-ice
data (HadISST; Rayner et al., 2003), and historical anthropogenic (carbon dioxide,
methane, and ozone) and natural (total solar irradiance) forcing factors. The simulation
covered the period of 1871–2007 (137 years).
Although the simulation period included recent times covered by observational data,
we assumed that the only variable that could be obtained was the annual mean of $\delta^{18}O$ in
the proxies. We based this assumption on the fact that we wished to perform the DA for a
period in which no direct measurements were available, and there were only climate
proxies covering the period. Therefore, the temporal resolutions of the "observations" and
"simulations" were also annual, considering the typical temporal resolution of the proxies.
Observations were generated by adding Gaussian noise to the truth. The spatial
distribution of the observations mimicked that of the proxies. The spatial distributions of
each proxy for various periods are mapped in Figure 1. As can be seen from the figure,
the distributions and the number of proxies varied with time. However, for the sake of
simplicity, the distributions of the proxies were assumed to be constant over time in the
CTRL experiment (Figure 1 a). The size of the observation errors will be discussed in
Section 2.4.
The state vector consisted of five variables; surface air temperature and amount of
precipitation, as well as the isotopic composition in precipitation, coral, and tree-ring
cellulose. The first three variables were obtained from the isotope-enabled GCM, and the
other two variables were obtained from the proxy models driven by the output of the
GCM.

**2.3.2.    Real proxy data assimilation**
The second (REAL) experiment assimilated proxy data sampled in the real world. To
mimic realistic conditions, SST and sea-ice concentration data to be used as model forcing
were modified from observational to modeled data. In reality, there were no direct
observations available for the target period of the proxy DA. Therefore, to reliably
evaluate the feasibility of proxy DA, the first estimate should be constructed using
modeled SST, as opposed to observed SST. We used SST data from the historical run of
the Coupled Model Intercomparison Project Phase 5 (CMIP5; Taylor et al., 2007) from
the atmosphere-ocean coupled version of MIROC5 (Watanabe et al., 2010) obtained from
the CMIP5 data server (https://pcmdi.llnl.gov/search/cmip5/).
Because the experiment was not an OSSE, nature run was not necessary.

### 259 2.3.3.    Sensitivity experiments

Four sensitivity experiments were conducted to test the robustness of the results of
the proxy DA. In the first sensitivity experiment (CGCM), the simulation run was
constructed from the simulation forced by the modeled SST and sea ice as in the REAL
experiment. The other settings for the simulation run were the same as those in the CTRL
experiment. The nature run was the same as that of the CTRL experiment. Thus, this
experiment investigated how the reconstruction skill of the results was decreased by using
the simulated SST compared to the CTRL.
In the second sensitivity experiment (VOBS), the experimental design was the same
as that in the CGCM, except for the number of proxies that were assimilated. In the
CGCM experiment, the distribution and number of proxies were set to be constant over
time, as in the CTRL experiment. In the VOBS experiment, the distribution and number
of proxies varied with time. Thus, this experiment investigated how the reconstruction
skill was decreased by changing the number of proxies compared to the CGCM.

In the third sensitivity experiment (T2-Assim), reconstructed surface temperature ($T_r$)

was assimilated. The purpose of the experiment was to compare the skill of the
reconstructed DA with that of the proxy DA. The experimental design was the same as
that in the CTRL experiment, except for the variables that were assimilated. The
reconstructed temperature was generated with a linear regression model of $T_r = a +$
$b \times \delta^{18}O$ where a and b are coefficients and $\delta^{18}O$ is the observed isotope ratio. The
coefficients are calibrated with the observed isotope ratio and the true temperature in the
CTRL for the period of 1871 to 1950 (80 years). If the correlation between the isotope
ratio and the temperature during the calibration period was not statistically significant ($p$
$< 0.10$), the data was discarded following Mann et al. (2008). This screening process
reduced the available data from 94 to 81 grid points.

The final sensitivity (M08) experiment was used to examine the sensitivity to the

observation network. The experimental design was the same as for the CTRL, except for
the spatial distribution of the proxy. The proxy network used in the experiment was the
same as that of Mann et al. (2008). We assumed that isotopic information was available
for all the sites, even when this was not the case. For example, even if only tree-ring width
data were available at some of the sites in Mann et al. (2008), in this experiment we
assumed that isotopic data recorded in tree-ring cellulose were available at the site. The
number of grids containing observations were 94 and 250 for the CTRL experiment and
M08 respectively. The T2-Assim and the M08 were compared with CTRL.
The experimental designs are summarized in Table 1.

## 2.4.    Observation data

We used paleoclimate data archived at the National Oceanic and Atmospheric
Administration (NOAA; https://www.ncdc.noaa.gov/data-access/paleoclimatology-data)
and data used in the PAGES 2k Consortium (2013). Additionally, 22 tree-ring cellulose
and 7 ice core data sets were collected separately from published papers. We only used
oxygen isotopic data ($^{18}$O) whose temporal resolution was higher than annual; proxies
whose resolution was lower than annual were excluded. The full list of proxies used in
this study is given in the Appendix. Following Crespin et al. (2009) and Goosse et al.
(2010), all proxy records were first normalized, and then averaged onto a T42 grid box to
eliminate model bias and produce a regional grid box composite. To compare the results
from each experiment effectively, the assimilated variables were all normalized in both
the simulation and nature runs, and in the observations in all the experiments.
Errors were added to the truth in a normalized manner to provide the observation for
all the experiment other than REAL. The normalized error was uniformly set to 0.50 for
all the proxies. This was based on the measurement error of $\delta^{18}O$ in ice cores being
reported to range from 0.05 to 0.2‰ (e.g., Rhodes et al., 2012; Takeuchi et al., 2014), and
the corresponding normalized error (measurement error divided by standard deviation of
proxy) then ranges from 0.03 to 0.1, with an average of 0.06. Similarly, the measurement
error of $\delta^{18}O$ in coral ranges from 0.03 to 0.11‰ (e.g., Asami et al., 2004; Goodkin et al.,
2008), and the corresponding normalized error ranges from 0.24 to 1.1, with an average
of 0.53. The measurement error of $\delta^{18}O$ in tree-ring cellulose ranges from 0.1 to 0.3‰
(e.g., Managave et al, 2011; Young et al, 2015), and the corresponding normalized error
ranges from 0.08 to 0.55, with an average of 0.28. In practice, due to the error of
representativeness and that in observation operator, it is common to increase the
observation errors to ensure that the analysis functions effectively (Yoshimura et al.,
2014). Furthermore, the measurement errors were not always available; therefore, a
uniform value of 0.5 was used for all the proxies. The corresponding signal-to-noise ratio
(SNR) is 2.0. The errors are assumed to be independent for all the experiments.

**3.     Results from the OSSE**

The time series of the first estimation, the analysis, and the real values for $\delta^{18}O$ in
corals are compared as an example in Figure 2 at a location where observational data were
available (1°N, 157°W). Because the first estimate was the same for all reconstruction
years, it is drawn as horizontal lines. After the assimilation, the analysis agreed well with
the real values (R = 0.96, $p < 0.001$). This confirmed that the assimilation performed well.
We then examined how accurately the other variables were reconstructed by assimilating
isotopic information. Figure 2 also shows the time series of surface air temperature and
precipitation for the same site. There was a clear agreement between the analysis and the
truth for both variables (R = 0.92 and 0.88 respectively for temperature and precipitation).
This indicated that temperature and precipitation were effectively reconstructed by
assimilating isotopic variables at this site. This was because the isotope ratio in corals has
a signature not only from temperature as given in Eq. 1, but also precipitation (Liu et al.,
2013); the correlation with $\delta^{18}O_{coral}$ was -0.88 ($p$ <0.001) for both temperature and
precipitation, respectively. This example shows that the isotopic proxy records more than
one variable.
Figure 3 maps the correlation coefficients between the analysis and the truth for the
isotope ratio, temperature, and precipitation for 1970–1999. Because the first estimate
was constant over time, the temporal correlation between the first estimate and the real
value was zero everywhere. Thus, a positive correlation indicated that the DA improved
the simulation.

The correlation for $\delta^{18}O$ in precipitation were high at the observation sites, regardless

of the proxy type. This was because $\delta^{18}O$ in both corals and trees is affected by the isotopic
composition in precipitated water derived from sea water or soil water. The correlation
for $\delta^{18}O$ in tree-ring cellulose were also high at the observation sites. On the other hand,
the high correlation for $\delta^{18}O$ in corals were not limited around the observation sites but
were generally high at low- to mid-latitudes. Similarly, the correlation was high at low-
to mid-latitudes for surface temperature. The correlation was also statistically significant
($p < 0.05$) around the observation sites in high latitude. In contrast, closely correlated
areas were restricted to low-latitude for precipitation.

How can the spatial distribution of the correlation pattern be explained; i.e., what do

the proxies represent? To investigate this question, empirical orthogonal function (EOF)
analysis was conducted for the simulated $\delta^{18}O$ in precipitation, corals, and tree-ring
cellulose. Only grids that contained observations were included in the analysis. The
variables were centered around their means before the analysis. The data covered the
period 1871–2007. The EOF patterns and temporal correlations between surface
temperature and the characteristic evolution of EOF, or the principal components (PCs)
of the first mode of each proxy are shown in Figure 4.

The first mode of $\delta^{18}O$ in ice core explains 14.3% of the total variance ant it is the

only significant mode according to the Rule of Thumb (North et al., 1982) (the first and
the second mode were indistinguishable). The maximum loadings were in Greenland and
Antarctica where temperature increase has been observed for the past hundred years (e.g.
Hartmann et al., 2013). Indeed, the PC1 shows the significant trend and is correlated with
global mean surface temperature (R=0.44, $p < 0.001$). Therefore, it is legitimate to regard
ice core data as a proxy of global temperature as revealed from observation (Schneider
and Noone, 2007).

The first modes of $\delta^{18}O$ in corals, and tree-ring cellulose represent ENSO. The

explained variance of the first modes of $\delta^{18}O$ in corals, and tree-ring cellulose was 44.2,
and 19.0%, respectively. The maximum loadings occurred in the central Pacific for corals,
and Tibet for tree-ring cellulose. The temporal correlation between the PC1s and NINO3
index were 0.95, and 0.37 for corals and tree-ring cellulose, respectively. Because the
isotopic composition in corals is influenced by sea temperature, it is expected that the
$\delta^{18}O$ in corals from the central Pacific records the ENSO signature. Interestingly, the
analysis revealed that the $\delta^{18}O$ in tree-ring cellulose was also influenced by ENSO; hence,
this proxy contributes to the reconstruction of temperature and precipitation over the
tropical Pacific. Indeed, many previous studies have reported the link between $\delta^{18}O$ in
tree-ring cellulose and ENSO (Sano et al. 2012; Xu et al. 2011; 2013; 2015). Xu et al.
(2011) inferred the link is caused by the association between ENSO and Indian monsoon
rainfall (e.g. Rasmusson and Carpenter, 1983). The positive phase of ENSO results in a
decrease in summer monsoon rainfall in India, which leads to dry conditions in summer.
The decrease in precipitation leads to isotopically-enriched precipitation, and the dry
conditions enhance the enrichment of water in leaves. Correspondingly, the $\delta^{18}O$ in tree-
ring cellulose becomes heavier than normal in the positive phase of ENSO. Due to the
relationships between the coral and tree-ring cellulose data and ENSO, the correlation
coefficient between the analysis and the truth for the NINO3 index was as high as 0.95 ($p$
$< 0.001$).

Although EOF analysis did not reveal any other significant correlation between PCs

and climate indices, climate indices for the North Atlantic Oscillation and Southern
Annular Mode calculated using the reconstructed data were significantly correlated with
the truth (0.59 and 0.46, respectively).

## 4.    Real proxy data assimilation

Based on the results of the idealized experiment described in the previous section, we performed a "real" proxy DA, in which sampled and measured data in the real world were assimilated.

The temporal correlation between the analysis and observations for temperature and precipitation are shown in Figure 5 (d, h). The observations were obtained from HadCRUT3 (Brohan et al., 2006) for temperature, and GHCN-Monthly Version 3 (Peterson and Vose, 1997) for precipitation.

Although the real proxy DA had reasonable skill, it was inferior relative to the CTRL experiment. We investigated the cause of the decreased skill using the outputs of the sensitivity experiments. The design of the experiments was changed in a stepwise fashion to more realistic conditions of proxy data assimilation from the idealized conditions. The correlations between the analysis and the truth, or the observation, for the experiments are shown in Figure 5. The truths for the CGCM and VOBS experiments were the same as those for the CTRL experiment. The global mean correlation coefficients for temperature, precipitation, and NINO3 in the experiments are summarized in Figure 6. Note that the correlation was averaged in the same domain for all the experiments to take into account the differences in representativeness.

In the CGCM experiment, the temporal correlations between the analysis and the truth
were similar to those in the CTRL experiment for both temperature and precipitation
(Figure 5 b, f). This indicates that ENSO and its impacts were well represented in the
modeled SST used to construct the "simulation". Watanabe et al. (2010) reported similar
modeled SST and observational values for the amplitude of ENSO measured by the
NINO3 index, and the spatial patterns of the temperature and precipitation fields
regressed on the NINO3 time series (see Figures 13 and 14 in their report).
Because the number of proxies for assimilation differed from that in the CGCM
experiment, it was not straightforward to compare the results of the REAL experiment
with those of the CGCM experiment. To enable an effective comparison of the results,
the same number of proxies were assimilated in the VOBS experiment as in the REAL
experiment and the same settings were used as in the CGCM experiment for the other
variables. Consequently, the performance of the assimilation of the VOBS experiment
was similar to that of the CGCM experiment for 1970–1999.
When the REAL and VOBS experiments were compared, the correlation coefficients
for temperature were significantly decreased over the Indian Ocean, eastern Pacific, and
Atlantic Ocean. These areas corresponded to areas of low reproducibility in the coral
model (Liu et al, 2014). The effects of sea current and river flow in these areas, which
were not included in the coral model, were deemed to be considerable. Although we
cannot attribute all the decreased skill to the coral model, the reproducibility of $\delta^{18}O$ in
corals in these areas requires improvement to enhance the performance of the assimilation.

**5.        Discussion**
**5.1.        Comparison with the reconstructed temperature assimilation**

Hughes and Ammann (2009) recommended assimilating measured proxy data, as

opposed to reconstructed data derived from the proxy data. This subsection compares the
results from the CTRL and T2-Assim experiments.

Figure 7 shows the spatial distribution of the correlation coefficients for temperature

and precipitation between the truth and the analysis for each experiment. As a whole, the
reconstruction skill was slightly degraded in T2-Assim compared with CTRL with the
global mean correlation coefficients for temperature (precipitation) of 0.50 (0.30), 0.45
(0.23), for CTRL and T2-Assim, respectively. On the other hand, the skill of proxy DA
was not always better than that of T2-Assim (e.g. temperature in tropical Atlantic Ocean).
Those pros and cons can be explained by the difference in the observation error and the
structure of Kalman gain. Figure 8 shows the SNR of the $T_r$ ranging from 0.22 to 1.6 with
the average of 0.65. Accordingly, the observation error is larger than that of CTRL
everywhere, and this resulted in the reduction of the reconstruction skill. On the other
hand, the better skill in T2-Assim should be owing to the difference in Kalman gain. The
Kalman gain determines analysis increments by spreading the information in observations
through the covariance between the prior and the prior-estimated observations. We found
that the correlations between the prior (temperature) and the prior-estimated observation
(temperature and $\delta^{18}O$ for T2-Assim and CTRL, respectively) were consistently higher in
T2-Assim than in CTRL (not shown) as Dee et al. (2016) showed. Thus, the information
in the observations were more effectively spread to the analysis in T2-Assim, and this
resulted in the improved skill. Note that the screening process hardly hampered the
reconstruction skill, because even if the reconstructed temperature was fully used (i.e. not
screened), the skill was almost the same as T2-Assim.
Conducting similar experiments, Dee et al. (2016) also concluded that the
reconstruction skill was almost the same among proxy DA and reconstructed DA if the
relation between the reconstructed variable and the proxy is linear. As isotope-enabled
GCMs (Schmidt et al. 2007; LeGrande and Schmidt. 2009) and observations and models
for tree-rings width (D'Arrigo et al. 2008; Evans et al. 2014; Dee et al., 2016) have
demonstrated, however, the relations between the proxies and climate are non-linear and
non-stationary as well. Thus, it is difficult to expect that the skill of reconstructed DA will
be the same as that of proxy DA if we have the well-defined forward proxy models
(Hughes and Ammann, 2009). Although the current models are far from perfect as
implicated in Sect. 4.2, the assimilation of proxy data will offer a useful tool for the
reconstruction of paleoclimate, in which the relationship between the proxies and climate
constructed with the present-day conditions does not apply.

**5.2.       Sensitivity to the distribution of the proxies**

The skill of the proxy DA was relatively low over Eurasia and North America, even
in the idealized experiment. It was unclear whether this was because of limitations in the
proxy data assimilation or the scant distribution of the proxies. This subsection
investigates the reasons for the relatively low reproducibility in these areas by comparing
the results of the CTRL and M08 experiments, focusing on North America. The number
of grids for which proxy data were available over North America was 11 and 126 for the
CTRL and M08, respectively.
The results for North America are shown in Figure 9. The figure shows the temporal
correlation coefficients between the analysis and the truth for surface air temperature and
precipitation. The correlation coefficients were calculated for 1970–1999. The skill was
high in the area in which the proxies were densely distributed for both variables. The
values of the coefficients averaged over the United States (30–50°N, 80–120°W) were
0.69 and 0.58 for temperature and precipitation, respectively. Compared to the
coefficients of 0.23 and 0.21, respectively, in the CTRL experiment, the skill was
enhanced for both variables. This implies that the performance of the reconstruction was
strongly dependent on the distribution of the proxy data. Taking into consideration that
proxy DA can assimilate not only proxy data but also reconstructed data, proxy DA can
take advantage of the use of increasingly large amounts of data. Although it is beyond the
scope of this study, the combined use of these data is expected to improve the performance
of proxy DA.

**6.     Conclusion and summary**
The feasibility of using proxy DA for paleoclimate reconstruction was examined in
both idealized and real conditions experiments. The idealized (CTRL) experiment had
high skill at low latitudes due to the dependency of coral data on temperature and
precipitation in these regions, and the correlation between ENSO and $\delta^{18}$O in corals in
Pacific and tree-ring cellulose in Tibet. Encouraged by the results, real proxy DA was
performed, where the simulation run was constructed from the simulation forced by the
modeled SST, and the real (observed) proxy data were assimilated into the simulation
(REAL experiment). The skill of the reconstruction decreased compared to CTRL.

To investigate the reason for the relatively low skill in REAL compared to CTRL, we

performed additional experiments; CGCM and VOBS. The imperfect SST used to drive
the CGCM experiment resulted in a slight reduction of the skill compared to the CTRL
experiment with perfect SST. This was because ENSO, which is the most important mode
for the reconstruction, was well represented in the modeled SST. The result is encouraging
because to apply the DA system to reconstruct ages where no instrumental observation is
available, we must rely on SST simulated by coupled GCM. Similarly, assimilating the
unfixed number of the observation only slightly decreased the reconstruction skill as
shown in the comparison between CGCM and VOBS.

From the suite of experiments, more than half of the difference between CTRL and

REAL remained unexplained. This remaining difference can have a lot of origins: e.g.
errors in the isotope incorporated atmospheric GCM, the proxy models, the proxy data
and so on. The errors in the models include such as model biases and missing or overly
simplified model components. For instance, the coral model does not take into account
the impact of ocean current or river runoff in this study. Furthermore, post-depositional
processes for simulating isotope ratio in ice core were not included at all. Those processes
should be included to enable more efficient utilization of all the data. The errors in proxy
data include such as misrepresentation of the targeted temporal and/or spatial scales. It is
also possible that the data were highly distorted by non-climatic factors. Thus, a thorough
quality control, similar to the procedures used in weather forecasting, should be
conducted before assimilation (e.g. Appendix B of Compo et al., 2011). At this stage, it
is difficult to show the relative contributions of each factor to the degraded skill in REAL,
it is necessary to estimate the impact of structural errors in models as done in Dee et al.

(2016).

Although the skill of proxy DA is dependent on the reproducibility of the models and
the number and quality of the observations, the results suggest that it is feasible to
constrain climate using only proxies. Especially, ENSO and ENSO-related variations in
temperature and precipitation should be reliably reconstructed even with the current
proxy DA system and proxy network used in this study because the correlation coefficient
between the analysis and the observations was as high as 0.83 in the REAL experiment.
Although the reconstruction of ENSO is dependent on data from corals, and the time span
covered by corals is relatively short (a few hundred years), ENSO can still be reliably
reconstructed due to its global impact, as was demonstrated in the relationship between
isotopes in tree-ring cellulose from Tibet.
Moreover, we expect that the reproducibility will increase as more proxy data become
available because it was heavily dependent on the spatial distribution. Given that proxy
DA can assimilate both proxy data and data reconstructed from proxy, and that the
reconstruction skill in reconstructed DA is partly superior to proxy DA, the combined use
of the two types of data is beneficial for the performance. In that case, care must be taken
not to assimilate dependent information (e.g. proxy data and reconstructed data from the
same proxy).

The DA algorithm used in this study did not consider non-stationarity among proxies

and climate variables because the Kalman gain was constant over time. To address non-
stationarity, the Kalman gain for a specific reconstruction year should be constructed for
several tens of years before and after that year. Nevertheless, EnKF can only capture
linear relationships between observations and the modeled state. The use of other
algorithms, such as particle filter (e.g. van Leeuwen, 2009), or four-dimensional
variational assimilation (e.g. Rabier et al., 2000), should be investigated in future studies
for scenarios where non-linearity is not negligible. Thus, it is important in future studies
to investigate non-stationarity and non-linearity among proxies and climate variables to
identify suitable algorithms for proxy DA.

**7.      Acknowledgements**

The first author was supported by the Japan Society for the Promotion of Science (JSPS) via a Grant-in-Aid for JSPS Fellows. This study was supported by the Japan Society for the Promotion of Science Grants 15H01729, 26289160, and 23226012, the SOUSEI Program, the ArCS project of MEXT, Project S-12 of the Japanese Ministry of the Environment, and the CREST program of the Japan Science and Technology Agency.

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

**Tables**

**Table 1.** Experimental designs. The observation network used in the CTRL experiment is
denoted as Orig.

| | SST data to drive simulation run | SST data to drive truth run | Assimilated variable | Observation network | Missing data |
|---|---|---|---|---|---|
| CTRL | HadISST | HadISST | Simulated $\delta^{18}O$ | Orig | w/o missing |
| CGCM | Modeled SST | HadISST | Simulated $\delta^{18}O$ | Orig | w/o missing |
| VOBS | Modeled SST | HadISST | Simulated $\delta^{18}O$ | Orig | w/ missing |
| REAL | Modeled SST | - | Observed $\delta^{18}O$ | Orig | w/ missing |
| T2-Assim | HadISST | HadISST | Reconstructed T2 from simulated $\delta^{18}O$ | Orig | w/o missing |
| M08 | HadISST | HadISST | Simulated $\delta^{18}O$ | M08 | w/o missing |



**Figures**

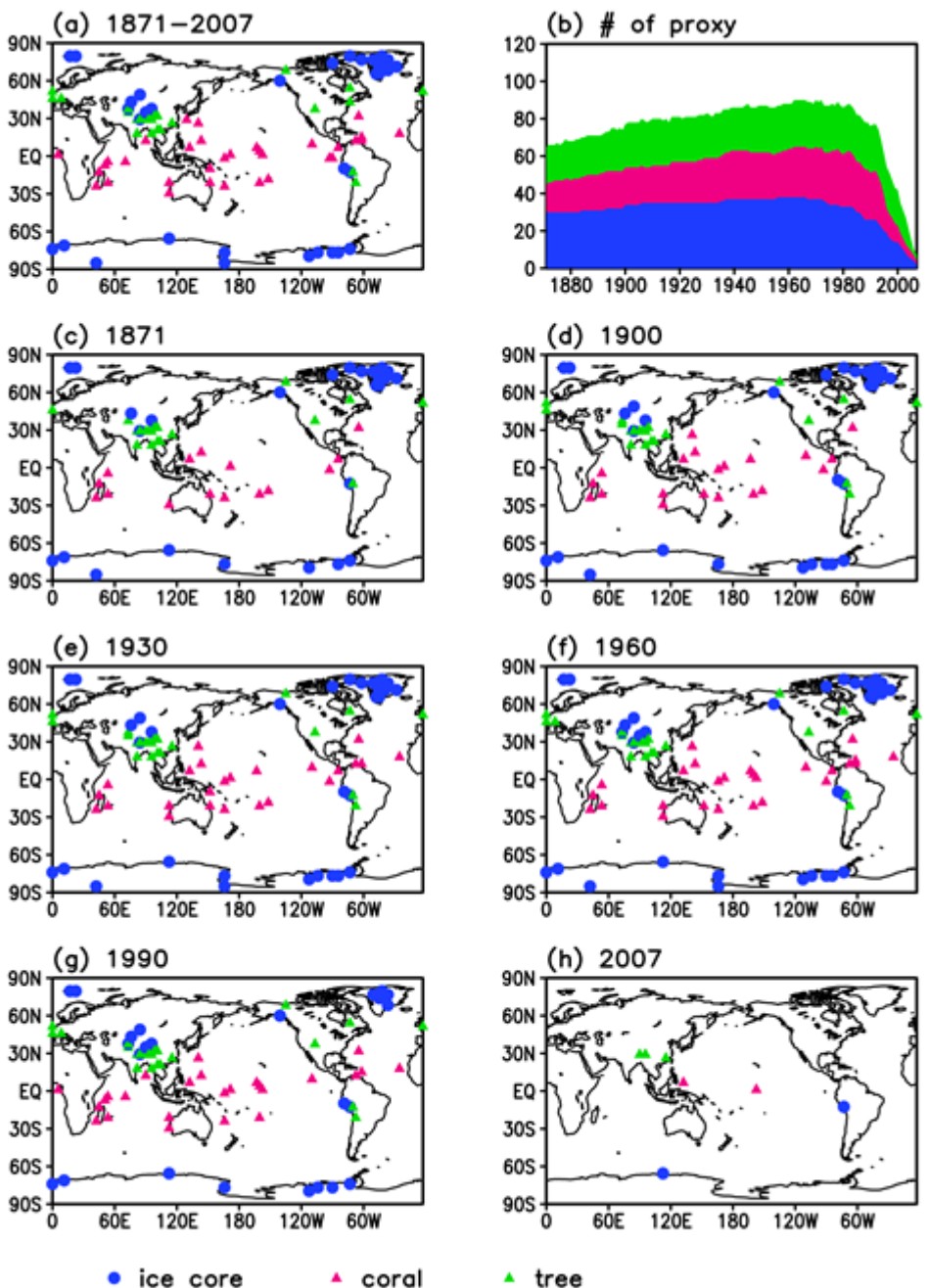

**Figure 1**

Spatial distribution of proxies ($\delta^{18}$O in ice cores, corals, and tree-ring cellulose, denoted by blue, pink, and green, respectively). (a) Proxies spanning at least one year during 1871–2000 are mapped (b) The number of proxies is depicted as a function of time. (c–

h) The spatial distributions of the proxies are mapped for (c) 1871, (d) 1900, (e) 1930, (f)
1960, (g) 1990, and (h) 2007.

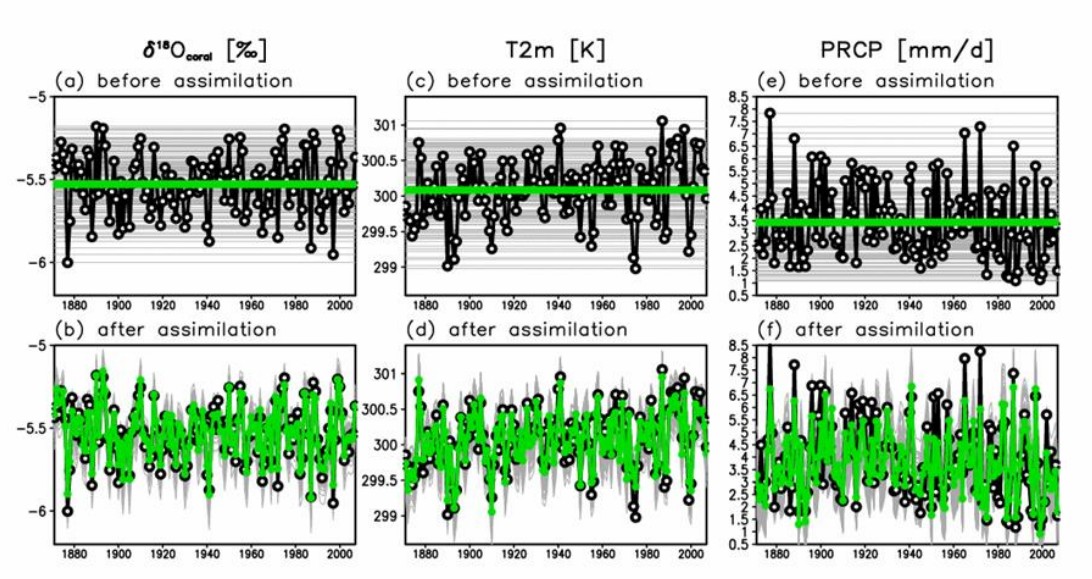


**Figure 2**
Annual mean $\delta^{18}O$ in corals at a location where observational data were available (1°N,
157°W) for (a) background and (b) analysis. The black line indicates the truth, gray lines
indicate ensemble members, and green line indicates the ensemble mean.

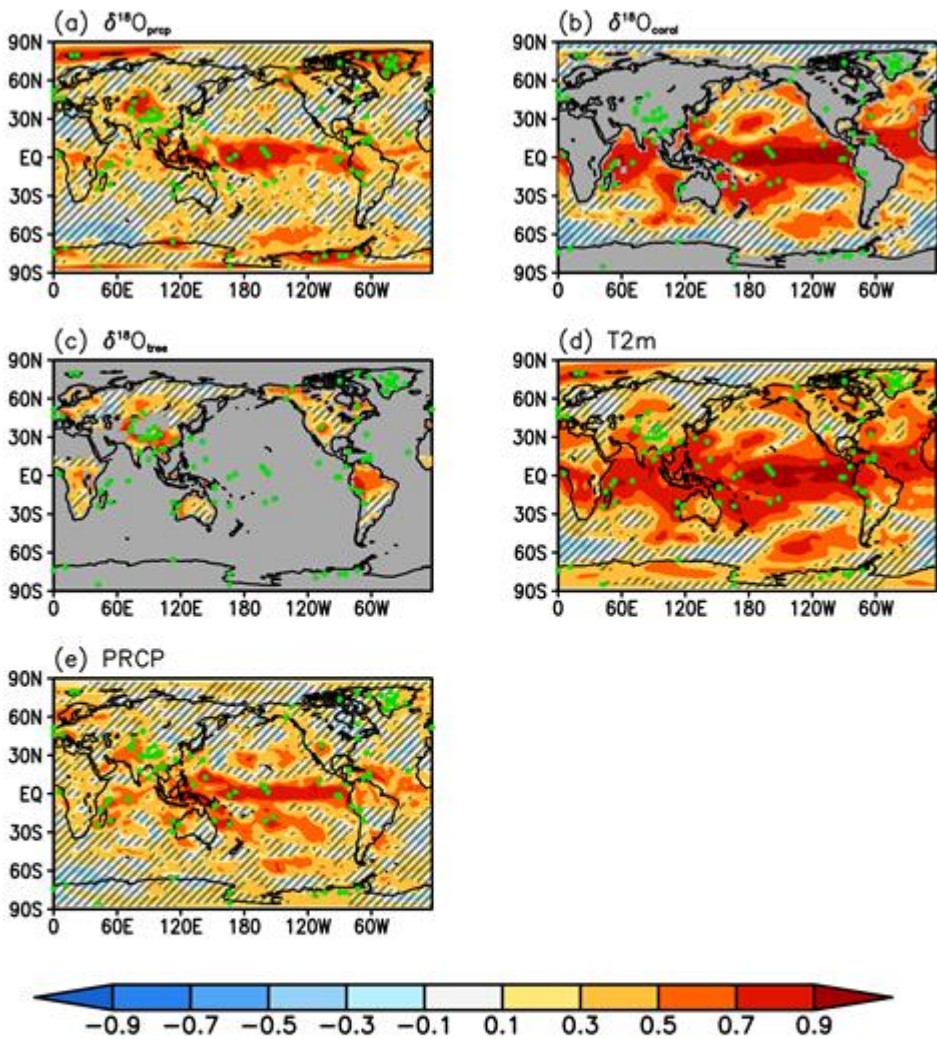

**Figure 3**
Temporal correlation between the analysis and the truth. The green dot represents the
location of the proxy sampling site. The hatched area indicates where the correlation is
not statistically significant ($p > 0.05$).

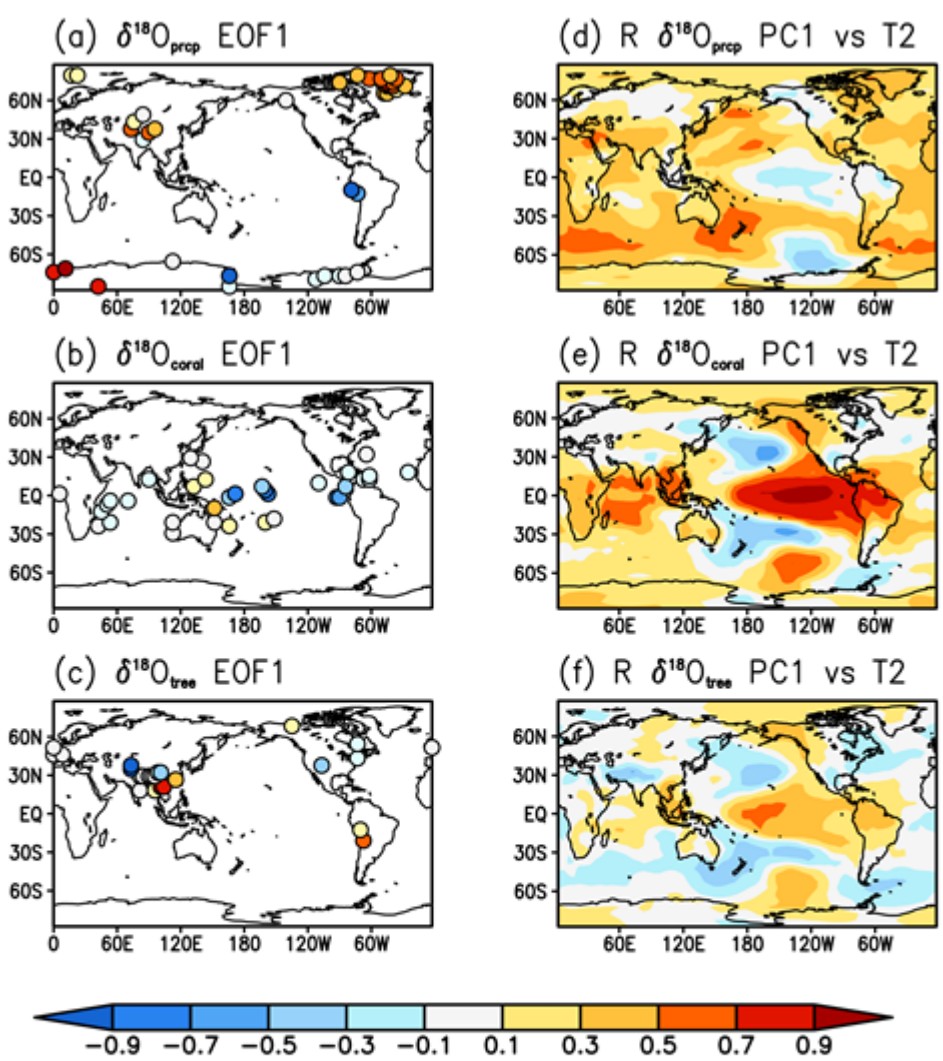

**Figure 4**
First mode of EOF and the correlation between PC1 and temperature for (a and d) ice
cores, (b and e) corals, and (c and f) tree-ring cellulose.

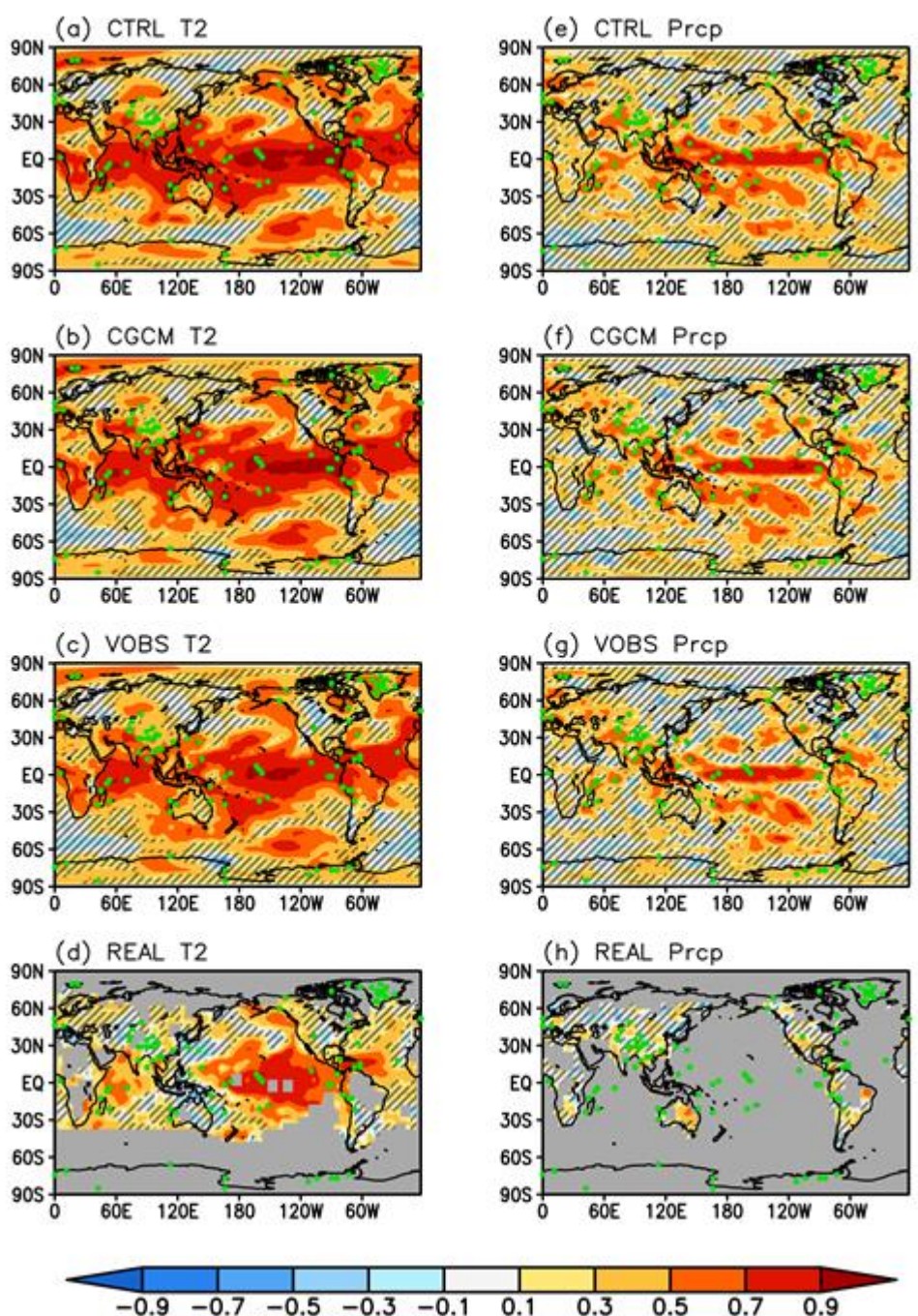


**Figure 5**

Temporal correlation between the analysis and the truth for (a–d) temperature and (e–h) precipitation, for each experiment. The green dot represents the location of the proxy sampling site. The hatched area indicates where the correlation is not statistically significant ($p > 0.05$).

817

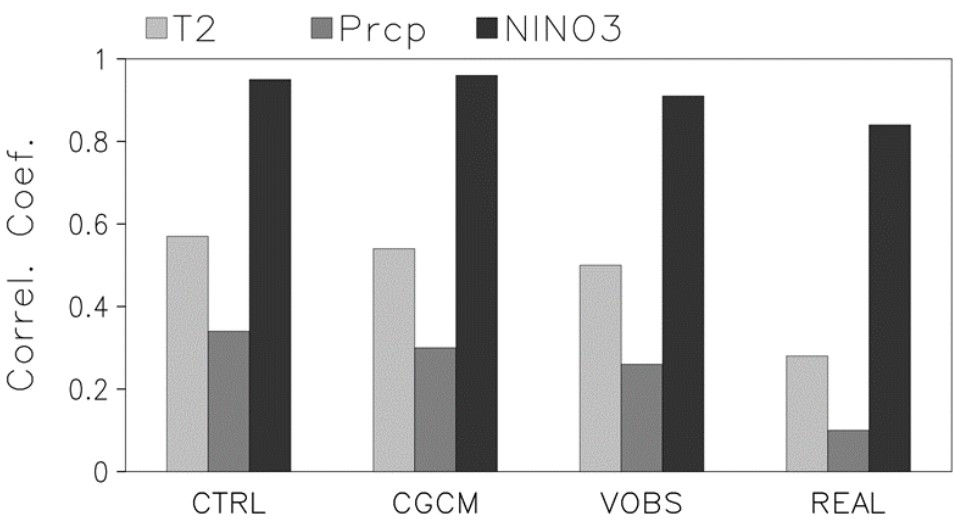

818

**Figure 6**

Temporal correlation between the analysis and the truth for each experiment for 1970–
1999. The values for temperature and precipitation are the global mean of the temporal
correlations.

823

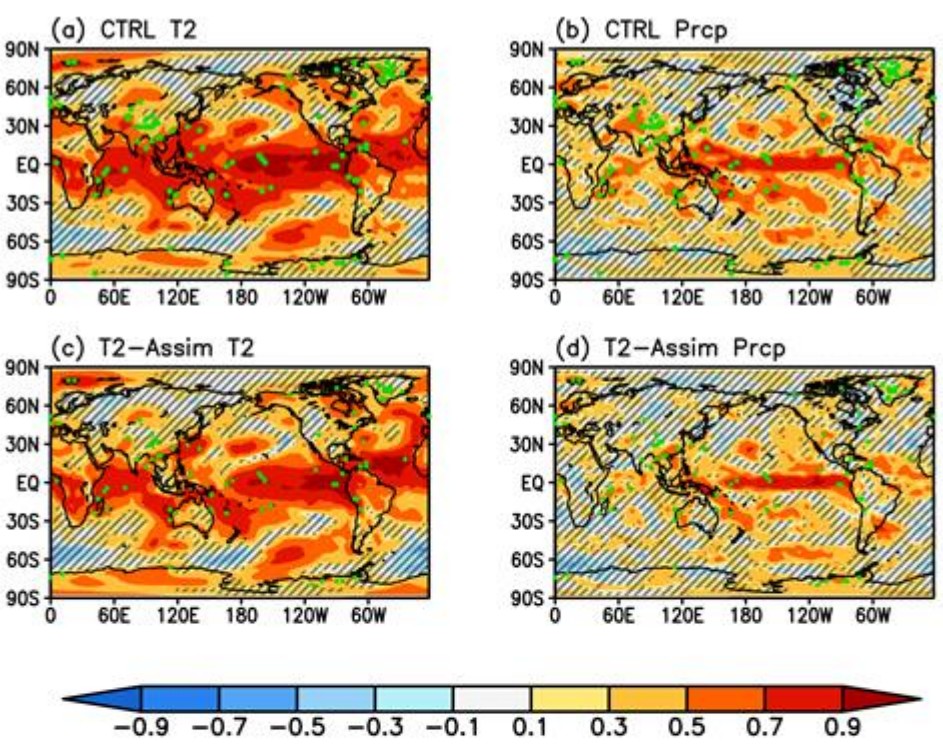

824

**Figure 7**

Temporal correlations between the analysis and the truth for (a, c) temperature and (b, d) precipitation, for (a, b) CTRL and (b, d) T2-Assim. The green dot represents the location of the proxy sampling site. The hatched area means that the correlation is not statistically significant ($p > 0.05$).


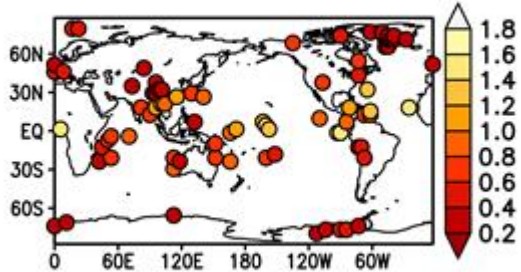


**Figure 8**

Signal to noise ratio (SNR) of the reconstructed temperature from the observation used

in CTRL.


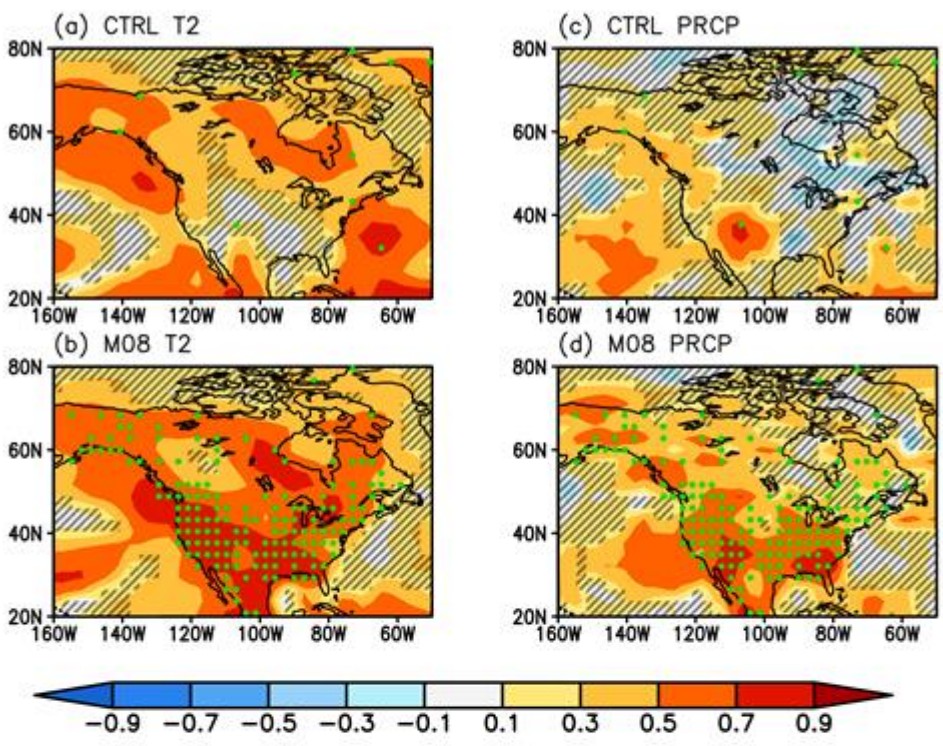

**Figure 9**

Temporal correlations in North America between the analysis and the truth for (a, b) temperature, and (c, d) precipitation, for experiments using different proxy networks. The green dot represents the location of the proxy sampling site. The hatched area indicates where the correlation is not statistically significant ($p > 0.05$).

842