# Peer review of "1. Introduction"

_Climate of the Past, 2016_

## Referee Comment (RC1) · Anonymous Referee #1 · 29 Nov 2016

Data assimilation in paleoclimatology is a rapidly growing field. The present paper addresses the model-data comparison step that is critical in every data assimilation scheme. Up to now, proxy records are generally first transformed to obtain a reconstruction of simulated variables such as temperature or precipitation before being assimilated. Simulating the measured quantity using proxy system models and performing the comparison directly for this variable provides in theory many advantages. The present study analyses those advantages and the potential limitations of the methodology based on both idealized and realistic experiments. It demonstrates the ability to directly assimilate isotopic composition of several proxies thanks to the application of forward proxy models. The study also identifies the regions/variables where the skill is

already satisfactory and the promising ways of improvement. The authors thus provide very interesting results for methodological developments and the application of data assimilation techniques in paleoclimatology. The study thus deserves publications in Climate of the Past but some modifications are required in the experimental design and in the discussion to reach conclusions that are easier to be interpreted and to be compared with recent work as detailed below.

General Points.

1/ Several groups are currently working on the direct assimilation of proxy records. The authors could not be blamed for not discussing all the very recent publications in the submitted version but a comparison of the conclusions reached here with the ones of Dee et al. (2016) must at least be included as the latter study is focused on a very close subject. In particular, Dee et al. (2016) compare a direct assimilation of isotopes using an isotope enabled atmospheric model with the assimilation of temperature derived from the proxy records, as in the present paper. The publication of those recent papers also requires to modify some sentences like lines 80-81 and 116-117 where it is said that it is the first time that proxy data are assimilated directly (see also Acevedo et al. 2016).

Acevedo W., B. Fallah, S. Reich, and U. Cubasch (2016). Assimilation of Pseudo-Tree-Ring-Width observations into an Atmospheric General Circulation Model. Clim. Past Discuss., doi:10.5194/cp-2016-92, 2016. Available at http://www.clim-past-discuss.net/cp-2016-92/

Dee, S.G., N.J. Steiger, J. Emile-Geay, and G.J. Hakim (2016): On the utility of proxy system modeling for estimating climate states over the Common Era. Journal of Advances in Modeling Earth Systems. doi:10.1002/2016MS000677. Available at http://onlinelibrary.wiley.com/doi/10.1002/2016MS000677/pdf

2/ I was surprised that the data assimilation method was not described at all in section 2.1. If I am right an ensemble Kalman filter is applied but this is only stated in the

conclusions (the word Kalman is mentioned first line 528). A long description of the method is not required but its main characteristics should at least be mentioned in section 2.1.

3/ The interpretation of experiment T2-ASSIM and its comparison with CTRL are not straightforward to me as the conclusions strongly depend on the signal to noise ratio selected and it is not possible from the information given in the paper to compare this signal to noise ratio with the error used in CTRL. One option would be to use the model results to estimate the impact of an error of 0.5 per mil on the isotopic composition, as imposed in CTRL, on a temperature reconstruction based on those isotopic records using simple statistical methods (for instance a regression as often done in paleoclimate reconstructions). Then, additional sensitivity experiments can be performed with such a temperature reconstruction derived from the isotopic composition (and not using the temperature simulated by the model) or alternatively assimilating temperature using the signal to noise ratio of this reconstruction that would be compatible with the error imposed in CTRL.

4/ The low skill of experiment REAL can have many origins: biases in climate models, limitations of proxy system models, non-climatic noise in the data, local signal in the records not represented in large-scale models, etc. The present study does not address the relative contribution of each of those elements and this is perfectly fine for me as it is not the goal of the present study. Nevertheless, some recommendations like line 51, line 497, line 502 , line 506 on the improvement of models seems relatively vague and not really justified by the results. I would thus recommend to be more careful and to focus on the main results of the study.

Specific points

1/ Abstract, line 42-43. This sentence is not clear without reading the main text. Please rephrase (see also general comment 2).

2/ Line 100. The data are not erroneous, this is the interpretation that is questionable.

3/ Line 143. The 'simplification' is valid for some variables but not for others that change more slowly such as oceanic temperatures.

4/ Line 150-151. What is meant by 'changing the algorithm'. The text should be more explicit and provide a reference if available.

5/ Line 176. A few words should be given on the version of MIROC5 applied as the reference is not available yet. In particular, it should be stated if only the atmospheric component is applied (as suggested lines 214-215) or if it is coupled to an interactive ocean.

6/ Line 189. Why is the deep ocean composition needed for corals that live in shallow waters?

7/ Line 250. I guess the four sensitivity experiments has to be compared to experiment CTRL. This should be already stated at this stage.

8/Line 322. Is it just a repetition of line 318 with a different sign or new information?

9/ Line 333. Why using 'on the other hand' here?

10/ Line 336. The results for temperature should be discussed too.

11/ Line 348. Is this increase noticed in simulation results or in observations? Please be more precise.

12/ Line 411-412. I would suppress this sentence as it does not bring new information.

13/ Line 415-419. I may miss something but I do not see how the low reproducibility of corals could play a role in the perfect model framework of CTRL as it is assumed that the climate and proxy models have no systematic bias (see also line 496).

---

## Referee Comment (RC2) · Anonymous Referee #2 · 20 Dec 2016

The authors present and analyze a novel approach to directly assimilate proxy information into GCM simulations to reconstruct past climate. They find that while assimilation of isotopic proxies is possible and is clearly beneficial in idealized simulations, the actual benefit of assimilating proxy data is limited due to model errors and the small number of assimilated proxies. Data assimilation in paleoclimatology has attracted a lot of attention recently and the science and methods are developing rapidly. This manuscript represents an important contribution to the field in that for one of the first times, proxy data (rather than reconstructed climatic variables) are assimilated directly for climate reconstructions. Therefore, I recommend this article to be published after the outstanding issues detailed below have been addressed.

[Figure]

General comments:

The sensitivity experiments conducted in this study only 'explain' a small fraction of the difference in correlation between the idealized setup (CTRL) and the application to real proxy data (REAL). The reasons for such a reduction in quality are manifold and include GCM model errors and errors in the proxy forward model that are not quantified in the current analysis. Proxy model errors are shortly discussed at the end of section 4, but it is not clear to me how one could attribute errors to the proxy model or the GCM in the absence of controlled experiments (as also stated by the authors in L504). While performing such controlled experiments with alternative proxy model / GCM combinations is clearly beyond the scope of this paper, I suggest the authors carefully reword the respective paragraphs.

In addition to trying to quantify the limitations of the current proxy DA setup by performing sensitivity experiments, the authors also try to answer a second question: namely whether direct assimilation of proxy data is superior to assimilating climatic variables (here temperature) reconstructed from the proxy data. In contrast to the approach pursued here, it would seem easier to address this question using the REAL experimental setup. Based on this setup, one could derive reconstructed (gridded) temperature data from the exact same proxies that have been used in the REAL experiment and assimilate these reconstructed temperatures instead. Such an experimental framework would be instructive as to whether empirical proxy models (i.e. reconstructed temperatures) outperform the physics-based on-line proxy models. Alternatively, one could devise idealized experiments similar to the ones performed in the study in which one compares assimilations based on the assumption of a perfect proxy model. In contrast to the comparison presented here, one would need to compare the CTRL (or any other of the synthetic proxy experiments) to the corresponding experiment in which the proxy data (+ noise) from the truth run has been used to reconstruct temperatures which are then assimilated. Such analysis, however, may be beyond the scope of this paper and I would be perfectly happy if the authors decide to focus on the main message of

the manuscript – the proxy data assimilation and partial attribution of its limited skill to quantifiable sources – only.

The data assimilation method is not described at all. Please add a short section on the data assimilation method with the relevant references. I suggest to focus on the choices and setup specific to this study and to provide the appropriate references; an in-depth introduction to the data assimilation method would only be needed if you chose a non-standard assimilation method that is not documented elsewhere. If, as suggested by the final paragraph of the manuscript, an EnKF has been used, then I suggest to also analyse the spread to error ratio or compute rank histograms to get an impression whether the analysis spread matches the analysis error and the analysis is well calibrated. Lack of calibration (usually overconfidence) is likely due to a misrepresentation of the observation error matrix (either underestimation of observation error or correlated errors).

Use of the term 'accuracy': The authors repeatedly use the term 'accuracy' to describe the quality of the analysis. This use of language is somewhat misleading, as accuracy in forecast verification has a specific meaning and the appropriate verification score to measure accuracy would be the mean squared or mean absolute error, whereas the correlation is a measure of forecast / analysis association (e.g. Murphy, 1993). I suggest to either rephrase and write of "improved assimilation", "enhanced correlation" etc. or to clearly state that accuracy refers to correlation throughout the manuscript.

Specific comments:

L112: This issue seems important and I think it would be worth revisiting in the conclusions.

L267: stemming from

L363-365/7: Is this a direct quote from the Xu et al. paper? If so I suggest labelling this as such by using quotation marks.

L385: for precipitation

L440: slightly more accurately?

L487ff: if the only difference in simulations is observed vs. simulated SSTs, I suggest the authors refrain from using the term forcing in the following lines for better readability.

L499ff: The discussion of the differences of the various sensitivity experiments is hard to read. I suggest to streamline and reword this section along the lines of "Imperfect SST used to drive the CGCM simulation resulted in a slight reduction of correlation compared to the CTRL experiment with perfect SST."

L513: non-climatic factors.

L514: add reference, e.g. Appendix B of Compo et al. 2011

L525: I suggest to mention that not in all cases direct proxy DA will be beneficial compared to assimilating empirically reconstructed variables. Also, while assimilating more data is expected to increase the quality of the analysis, care has to be taken in assimilating dependent information (e.g. direct assimilation of proxy data and reconstructed variables derived from the same proxy data).

Figure 4: The figure labels denote EOF2 whereas only EOF1 is mentioned in the text. Please fix.

---

## Editor Comment (EC1) · V. Rath (Editor) · 18 Jan 2017

Dear Authors, you study has received two positive reviews, and you are required now to reply to the referees in detail, in order to conclude the discussion phase. I invite you to submit a revised manuscript to CP, following the suggestions made by the referees. Best regards, Volker Rath

---

## Author Response (AR1)

Dear Dr. Volker Rath,

We would like to submit our revised manuscript of cp-2016-121.

We believe the paper has been improved very much by taking account all the comments that two reviewers kindly gave to us. Even though we have changed many parts of the manuscript, our final messages remain unchanged.

Hereafter, point-by-point responses to all the reviewer's comments follow. The reviewer's comments in blue italic font, the replies are in black and the changes in red with the line number of the marked-up version.

Best regards,

Atsushi Okazaki and Kei Yoshimura

For Anonymous Referee #1

Note that some of the replies are used in common, since comments 2, 3, 4 are similar to the comments 3, 2, 1 by the anonymous referee #2.

*Data assimilation in paleoclimatology is a rapidly growing field. The present paper addresses the model-data comparison step that is critical in every data assimilation scheme. Up to now, proxy records are generally first transformed to obtain a reconstruction of simulated variables such as temperature or precipitation before being assimilated. Simulating the measured quantity using proxy system models and performing the comparison directly for this variable provides in theory many advantages. The present study analyses those advantages and the potential limitations of the methodology based on both idealized and realistic experiments. It demonstrates the ability to directly assimilate isotopic composition of several proxies thanks to the application of forward proxy models. The study also identifies the regions/variables where the skill is already satisfactory and the promising ways of improvement. The authors thus provide very interesting results for methodological developments and the application of data assimilation techniques in paleoclimatology. The study thus deserves publications in Climate of the Past but some modifications are required in the experimental design and in the discussion to reach conclusions that are easier to be interpreted and to be compared with recent work as detailed below.*

Thank you very much for the positive and valuable comments.

1. *Several groups are currently working on the direct assimilation of proxy records. The authors could not be blamed for not discussing all the very recent publications in the submitted version but a comparison of the conclusions reached here with the ones of Dee et al. (2016) must at least be included as the latter study is focused on a very close subject. In particular, Dee et al. (2016) compare a direct assimilation of isotopes using an isotope enabled atmospheric model with the assimilation of temperature derived from the proxy records, as in the present paper. The publication of those recent papers also requires to modify some sentences like lines 80-81 and 116-117 where it is said that it is the first time that proxy data are assimilated directly (see also Acevedo et al. 2016).*

    *Acevedo W., B. Fallah, S. Reich, and U. Cubasch (2016). Assimilation of PseudoTree-Ring-Width observations into an Atmospheric General Circulation Model. Clim. Past Discuss., doi:10.5194/cp-2016-92, 2016. Available at*

*http://www.clim-pastdiscuss.net/cp-2016-92/*

*Dee, S.G., N.J. Steiger, J. Emile-Geay, and G.J. Hakim (2016): On the utility of proxy system modeling for estimating climate states over the Common Era. Journal of Advances in Modeling Earth Systems. doi:10.1002/2016MS000677. Available at http://onlinelibrary.wiley.com/doi/10.1002/2016MS000677/pdf*

We included the Acevedo et al. (2016) and Dee et al. (2016) in Sect. 1 and modified the corresponding sentences. Also, we included Dee et al. (2016) in Sect. 5.1 to discuss the comparison between proxy DA and reconstructed DA.

L85-87, L123-124, L126, L505-507, L595-598

2. *I was surprised that the data assimilation method was not described at all in section 2.1. If I am right an ensemble Kalman filter is applied but this is only stated in the conclusions (the word Kalman is mentioned first line 528). A long description of the method is not required but its main characteristics should at least be mentioned in section 2.1.*

The description of data assimilation method is included in the revised manuscript (L133-137). We used EnSRF (Whitaker and Mitchell, 2002) with slight modification following the previous studies (Bhend et al., 2012; Steiger et al., 2014).

L143-150, L155-163, L711-712, L829-830

3. *The interpretation of experiment T2-ASSIM and its comparison with CTRL are not straightforward to me as the conclusions strongly depend on the signal to noise ratio selected and it is not possible from the information given in the paper to compare this signal to noise ratio with the error used in CTRL. One option would be to use the model results to estimate the impact of an error of 0.5 per mil on the isotopic composition, as imposed in CTRL, on a temperature reconstruction based on those isotopic records using simple statistical methods (for instance a regression as often done in paleoclimate reconstructions). Then, additional sensitivity experiments can be performed with such a temperature reconstruction derived from the isotopic composition (and not using the temperature simulated by the model) or alternatively assimilating temperature using the signal to noise ratio of this reconstruction that would be compatible with the error imposed in CTRL.*

Thank you for the comments. We modified the experimental setting for T2-Assim following your suggestion. In the modified experiment, temperature is reconstructed from the isotopic records which is used in CTRL by simple regression-based method. Proxies whose correlation with local temperature during calibration period (1871-

1950) is not statistically significant (p < 0.10) are removed following Mann et al. (2008). This screening process reduced the available data from 94 to 81 grid points. Based on the correlation between isotope ratio and local temperature, SNR can be estimated through the equation (Mann et al., 2007):

$$\text{SNR} = \sqrt{\frac{r^2}{1 - r^2}}$$

where r is the correlation. SNR is shown in Fig. 8. Subsequently, this reconstructed temperature ($T_r$) is assimilated. The assimilated result is shown in Fig. 7. The result is slightly degraded in T2-Assim compared with CTRL due to relatively large error in $T_r$ (Fig. 8). As shown in Dee et al. (2016), the reconstruction skill is somewhat compensated by the structure of Kalman gain. Figure S1 shows the correlation scale length to show the difference in the structure between CTRL and T2-Assim. The correlation scale length was found by computing point correlation between the prior (temperature) and the prior-estimated observation (temperature and $\delta^{18}O$ for T2-Assim and CTRL, respectively) for the observation grids, binning these correlations by distance, and computing the mean of each bin. The correlation is consistently high in T2-Assim, which means that the observation information is more effectively used to update the analysis. To sum up, the accuracies are not substantially different among proxy DA and reconstructed DA. However, we should note that this is only the case as long as the relation between temperature and isotope remain the same. L288-289, L292-301, L339-340, L461-518, Table 1, Figure 7, Figure 8

[Figure]

**Figure S 1** Mean correlation scale length for T2-Assim (red) and CTRL (blue). The prior is (a) temperature and (b) precipitation. The prior-estimated observation is temperature and $\delta^{18}O$ for T2-Assim and CTRL, respectively for the both panels.

4. *The low skill of experiment REAL can have many origins: biases in climate models, limitations of proxy system models, non-climatic noise in the data, local signal in the records not represented in large-scale models, etc. The present study does not address the relative contribution of each of those elements and this is perfectly fine for me as it is not the goal of the present study. Nevertheless, some recommendations like line 51, line 497, line 502, line 506 on the improvement of models seems relatively vague and not really justified by the results. I would thus recommend to be more careful and to focus on the main results of the study.*

   Thank you for the comments. We understand that there are multiple factors other than model errors for the low skill in REAL experiment and that we do not know their relative contribution. Thus, we carefully modified the abstract and Sect. 6 in which we avoided arguing vague explanation.
   L50-53, L455-456, L571-593, L595-598

*Specific points*

1. *Abstract, line 42-43. This sentence is not clear without reading the main text. Please rephrase (see also general comment 2).*

   Thank you for the comments. We omitted the sentence for better readability.

2. *Line 100. The data are not erroneous, this is the interpretation that is questionable.*

   We reworded that part as "such questionable reconstructed data". Thank you.
   L106

3. *Line 143. The 'simplification' is valid for some variables but not for others that change more slowly such as oceanic temperatures.*

   We clearly mentioned that the simplification is valid at least for atmospheric variables in the revised manuscript .
   L165-166

4. *Line 150-151. What is meant by 'changing the algorithm'. The text should be more explicit and provide a reference if available.*

   We rephrased the sentence as "the proxy DA could address non-stationarity if one uses temporally varying background ensemble".
L171-172, L623-624, L766-769, L816-817

5. *Line 176. A few words should be given on the version of MIROC5 applied as the reference is not available yet. In particular, it should be stated if only the atmospheric component is applied (as suggested lines 214-215) or if it is coupled to an interactive ocean.*

Thank you for the comments. The version of the model is five (hence MIROC"5") and we used only the atmospheric component of the GCM. To make it clearer, we changed the sentence as "we used a newly-developed model based on the atmospheric component of MIROC5".
L198

6. *Line 189. Why is the deep ocean composition needed for corals that live in shallow waters?*

Thank you for the comment. The isotopic ratio in the upper layer of the ocean is determined by the balance of precipitation, evaporation, and vertical mixing from deeper water, not deep water. We modified the term "deep" to "deeper" in the revised manuscript.
L211, L212

7. *Line 250. I guess the four sensitivity experiments has to be compared to experiment CTRL. This should be already stated at this stage.*

Two of them (i.e. CGCM and VOBS) were conducted to explain the difference among CTRL and REAL and the experimental settings were changed in a stepwise manner, from idealized way to more realistic way. Thus, CGCM were compared with CTRL, and VOBS were compared with CGCM. The other two were compared with CTRL. We included sentences explaining what experiment was used to evaluate each sensitivity experiment.
L279, L284-287, L310

8. *Line 322. Is it just a repetition of line 318 with a different sign or new information?*

No, it is not. The first sentence described the reconstruction skill for temperature and precipitation by comparing the analysis and the truth. On the other hand, the second sentence explained how the high reconstruction skill was achieved by comparing the assimilated variable ($\delta$) and the reconstructed variable (temperature and precipitation) at the site.

9. *Line 333. Why using 'on the other hand' here?*

The closely correlated area was limited around the observation site for $\delta^{18}O$ in tree-ring cellulose, but the high correlation was not limited around the observation site for $\delta^{18}O$ in coral. Thus, we used the "on the other hand" here. To make the context clearer, we modified the sentence in the revised manuscript (L345-350).

L367-369

10. *Line 336. The results for temperature should be discussed too.*

The results for temperature were included in the revised manuscript (L350-351). Thank you.

L369-370

11. *Line 348. Is this increase noticed in simulation results or in observations? Please be more precise.*

The temperature has been increased both in observations and simulations. In the manuscript, what we meant was observation. We modified the sentence and put a reference in the revised manuscript (L364-366).

L383-384, L701-707

12. *Line 411-412. I would suppress this sentence as it does not bring new information.*

Suppressed.

L448-450

13. *Line 415-419. I may miss something but I do not see how the low reproducibility of corals could play a role in the perfect model framework of CTRL as it is assumed that the climate and proxy models have no systematic bias (see also line 496).*

In this chapter, we compared VOBS and REAL, where VOBS is a perfect model experiment assuming that the climate and proxy models have no systematic bias and REAL is not a perfect model experiment. In the REAL, we assimilated observed data in the real world. Thus, models do have biases.

For Anonymous Referee #2

Note that some of the replies are used in common, since general comments 1, 2, 3 are similar to the comments 4, 3, 2 by the anonymous referee #1.

*The authors present and analyze a novel approach to directly assimilate proxy information into GCM simulations to reconstruct past climate. They find that while assimilation of isotopic proxies is possible and is clearly beneficial in idealized simulations, the actual benefit of assimilating proxy data is limited due to model errors and the small number of assimilated proxies. Data assimilation in paleoclimatology has attracted a lot of attention recently and the science and methods are developing rapidly. This manuscript represents an important contribution to the field in that for one of the first times, proxy data (rather than reconstructed climatic variables) are assimilated directly for climate reconstructions. Therefore, I recommend this article to be published after the outstanding issues detailed below have been addressed.*
Thank you very much for the positive and valuable comments.

*General comments:*

1. *The sensitivity experiments conducted in this study only 'explain' a small fraction of the difference in correlation between the idealized setup (CTRL) and the application to real proxy data (REAL). The reasons for such a reduction in quality are manifold and include GCM model errors and errors in the proxy forward model that are not quantified in the current analysis. Proxy model errors are shortly discussed at the end of section 4, but it is not clear to me how one could attribute errors to the proxy model or the GCM in the absence of controlled experiments (as also stated by the authors in L504). While performing such controlled experiments with alternative proxy model / GCM combinations is clearly beyond the scope of this paper, I suggest the authors carefully reword the respective paragraphs.*
Thank you for the comments. We understand that there are multiple factors other than model errors for the low skill in REAL experiment and that we do not know their relative contribution. Thus, we carefully modified the abstract and Sect. 6 in which we clearly mentioned that there remains a lot unexplained and avoided arguing that the model errors are the only reason for the degradation in REAL experiment.
L50-53, L455-456, L571-593, L595-598

2. *In addition to trying to quantify the limitations of the current proxy DA setup by performing sensitivity experiments, the authors also try to answer a second question: namely whether direct assimilation of proxy data is superior to assimilating climatic variables (here temperature) reconstructed from the proxy data. In contrast to the approach pursued here, it would seem easier to address this question using the REAL experimental setup. Based on this setup, one could derive reconstructed (gridded) temperature data from the exact same proxies that have been used in the REAL experiment and assimilate these reconstructed temperatures instead. Such an experimental framework would be instructive as to whether empirical proxy models (i.e. reconstructed temperatures) outperform the physics-based on-line proxy models. Alternatively, one could devise idealized experiments similar to the ones performed in the study in which one compares assimilations based on the assumption of a perfect proxy model. In contrast to the comparison presented here, one would need to compare the CTRL (or any other of the synthetic proxy experiments) to the corresponding experiment in which the proxy data (+ noise) from the truth run has been used to reconstruct temperatures which are then assimilated. Such analysis, however, may be beyond the scope of this paper and I would be perfectly happy if the authors decide to focus on the main message of the manuscript – the proxy data assimilation and partial attribution of its limited skill to quantifiable sources – only.*

Thank you for the comments. We modified the experimental setting for T2-Assim following your suggestion. In the modified experiment, temperature is reconstructed from the isotopic records which is used in CTRL by simple regression-based method. Proxies whose correlation with local temperature during calibration period (1871-1950) is not statistically significant ($p < 0.10$) are removed following Mann et al. (2008). This screening process reduced the available data from 94 to 81 grid points. Based on the correlation between isotope ratio and local temperature, SNR can be estimated through the equation (Mann et al., 2007):

$$\mathrm{SNR} = \sqrt{\frac{r^2}{1 - r^2}}$$

where r is the correlation. SNR is shown in Fig. 8. Subsequently, this reconstructed temperature ($T_r$) is assimilated. The assimilated result is shown in Fig. 7. The result is slightly degraded in T2-Assim compared with CTRL due to relatively large error in $T_r$ (Fig. 8). As shown in Dee et al. (2016), the reconstruction skill is somewhat compensated by the structure of Kalman gain. Figure S1 shows the correlation scale length to show the difference in the structure between CTRL and T2-Assim. The correlation scale length was found by computing point correlation between the prior (temperature) and the prior-estimated observation (temperature and $\delta^{18}O$ for T2-Assim and CTRL, respectively) for the observation grids, binning these correlations by distance, and computing the mean of each bin. The correlation is consistently high in T2-Assim, which means that the observation information is more effectively used to update the analysis. To sum up, the accuracies are not substantially different among proxy DA and reconstructed DA. However, we should note that this is only the case as long as the relation between temperature and isotope remain the same. L288-289, L292-301, L339-340, L461-518, Table 1, Figure 7, Figure 8

[Figure]

**Figure S 2** Mean correlation scale length for T2-Assim (red) and CTRL (blue). The prior is (a) temperature and (b) precipitation. The prior-estimated observation is temperature and $\delta^{18}O$ in coral for T2-Assim and CTRL, respectively for the both panels.

3. *The data assimilation method is not described at all. Please add a short section on the data assimilation method with the relevant references. I suggest to focus on the choices and setup specific to this study and to provide the appropriate references; an in-depth introduction to the data assimilation method would only be needed if you chose a non-standard assimilation method that is not documented elsewhere. If, as suggested by the final paragraph of the manuscript, an EnKF has been used, then I suggest to also analyse the spread to error ratio or compute rank histograms to get an impression whether the analysis spread matches the analysis error and the*

*analysis is well calibrated. Lack of calibration (usually overconfidence) is likely due to a misrepresentation of the observation error matrix (either underestimation of observation error or correlated errors).*

The description of data assimilation method is included in the revised manuscript (L133-137). We used EnSRF (Whitaker and Mitchell, 2001) with slight modification following the previous studies (Bhend et al., 2012; Steiger et al., 2014). As described in the manuscript, we did an offline approach, in which the analysis is not cycled to the simulation and the same background is used for every analysis step.

Figure S2 shows the spread and RMSE for surface temperature in REAL experiment. The posterior spread matches with RMSE for the first half of the period but it gradually diverges from the RMSE. This reflects the fact that the system has a difficulty in reconstructing temperature in mid- to high-latitude (Fig. 3), where temperature has been increasing in the period. However, the discrepancy does not necessarily mean that the system is not well calibrated. The relatively scarce observation and short correlation length scale must hamper the reproducibility there. On top of that, we speculate that the metrics such as spread-to-error ratio or rank histogram may not suit for the evaluation of the offline DA. In general, it takes several cycles for the spread to match with RMSE through improving the error covariance matrix in the online DA. Contrarily, because the offline DA uses the same background for every analysis step, the quality of the analysis error covariance remains the same (c.f. $\mathbf{P^a} = [\mathbf{I} - \mathbf{P^f}\mathbf{H^t}(\mathbf{HP^f}\mathbf{H} + \mathbf{R})^{-1}\mathbf{H}]\mathbf{P^f}$). Therefore, the spread will not tell how well the system is calibrated.

Instead, we show the sensitivity of the system to parameters (observation error and localization scale) in REAL experiment to show that the system is how optimal. The results show that the skill is moderately dependent on both the observation error and the localization scale. For the observation error, the results become better with larger error in the investigated range for the both variables. On the other hand, the sensitivity to the localization scale varies from variable to variable. For temperature, the correlation become better along with the scale in the investigated range. For precipitation, the localization scale of 12000km resulted in the best correlation.

The sensitivity is also different by which metric to be used. For instance, RMSE for precipitation becomes larger along with the observation error (not shown).

Given the results above and because the choice of the parameters does not change the main conclusion of the study, we keep using the original value of the parameters.
L143-150, L155-163, L711-712, L829-830

[Figure]

**Figure S 3** Global mean of the spread and RMSE for surface temperature in REAL. The spread of background and analysis and RMSE are shown in blue, red, and black, respectively.

[Figure]

**Figure S 4** Box-whisker plot of the distribution of all spatial values for the correlation for (a and b) temperature, and (c and d) precipitation in REAL. (a) and (c) shows the sensitivity to the observation error, and (b) and (d) shows the sensitivity to the localization scale.

4. *Use of the term 'accuracy': The authors repeatedly use the term 'accuracy' to describe the quality of the analysis. This use of language is somewhat misleading, as accuracy in forecast verification has a specific meaning and the appropriate verification score to measure accuracy would be the mean squared or mean absolute error, whereas the correlation is a measure of forecast / analysis association (e.g. Murphy, 1993). I suggest to either rephrase and write of "improved assimilation", "enhanced correlation" etc. or to clearly state that accuracy refers to correlation throughout the manuscript.*

Thank you for the comments. We used the term 'reconstruction skill' or 'skill' instead of 'accuracy' in the revised manuscript. In addition, we clearly stated that we use the correlation coefficient as a measurement of skill (L119-120).

L41, L47, L50, L107, L129-130, L278, L290, L424, L425, L521, L531, L546, L558, L599

*Specific comments:*

1. *L112: This issue seems important and I think it would be worth revisiting in the conclusions.*

   Thank you for the comment. We revisited the issue in the abstract and the conclusion in the revised manuscript (L48-50, L528-530).

   L54-56, L599-605

2. *L267: stemming from*

   The sentence was deleted because we changed the experimental design for T2-Assim.

3. *L363-365/7: Is this a direct quote from the Xu et al. paper? If so I suggest labelling this as such by using quotation marks.*

   No, it is not. We modified the sentence for better readability (L380-385).

   L398-403

4. *L385: for precipitation*

   Corrected.

   L423

5. *L440: slightly more accurately?*

   Section 5.1 were substantially modified following the general comment #2. Accordingly, the sentence was not used any more in the revised manuscript. Thank you.

6. *L487ff: if the only difference in simulations is observed vs. simulated SSTs, I suggest the authors refrain from using the term forcing in the following lines for better readability.*

We modified Sect. 6 significantly and the corresponding parts were omitted. Thank you.

7. *L499ff: The discussion of the differences of the various sensitivity experiments is hard to read. I suggest to streamline and reword this section along the lines of "Imperfect SST used to drive the CGCM simulation resulted in a slight reduction of correlation compared to the CTRL experiment with perfect SST."*

The corresponding sentences were rephrased in the revised manuscript following the suggestion (L505-507; 510-512). Thank you.

L563-564, L568-570

8. *L513: non-climatic factors.*

Corrected.

L593

9. *L514: add reference, e.g. Appendix B of Compo et al. 2011*

We added the reference (Appendix B of Compo et al., 2011) in the revised manuscript.

L595, L656-661

10. *L525: I suggest to mention that not in all cases direct proxy DA will be beneficial compared to assimilating empirically reconstructed variables. Also, while assimilating more data is expected to increase the quality of the analysis, care has to be taken in assimilating dependent information (e.g. direct assimilation of proxy data and reconstructed variables derived from the same proxy data).*

The both sentences were included in the revised manuscript (L444-445; 542-543; 544-546). Thank you.

L489-490, L610-617

11. *Figure 4: The figure labels denote EOF2 whereas only EOF1 is mentioned in the text. Please fix.*

Thank you for your pointing out. The figure was replaced with the correct figure.

Figure 4

[revised manuscript text omitted]

correlation coefficients for temperature (precipitation) were 0.49 (0.29), 0.50 (0.22), 0.39

(0.16), and 0.25 (0.10) for the experiments assimilating $\delta^{18}O$ in proxies, and those assimilating temperature with SNR values of 1.0, 0.50, and 0.25, respectively (Figure 8).

The values were higher for the assimilated $\delta^{18}O$ in proxy than for assimilated temperature, with SNR values of 0.25 and 0.50 for both precipitation and temperature. The temperature was reconstructed slightly accurately by assimilation of temperature with a low noise value (SNR = 1.0) than by assimilation of $\delta^{18}O$ in the proxies. Although using an SNR =

1.0 produced more accurate reconstructed field than the ordinal statistical reconstruction, the superior accuracy of the assimilation of proxy data relative to the assimilation of reconstructed temperature was dependent on the magnitude of the SNR; i.e., the accuracy of assimilation of the reconstructed values was dependent on the quality of the reconstructed data. The quality of the reconstructed data was in turn dependent on the stationarity between the proxies and climate, and the degree to which the proxy was affected by factors other than the variable of interest. As a whole, the reconstruction skill was slightly degraded in T2-Assim compared with CTRL with the global mean correlation coefficients for temperature (precipitation) of 0.50 (0.30), 0.45 (0.23), for

CTRL and T2-Assim, respectively. On the other hand, the skill of proxy DA was not always better than that of T2-Assim (e.g. temperature in tropical Atlantic Ocean). Those pros and cons can be explained by the difference in the observation error and the structure of Kalman gain. Figure 8 shows the SNR of the $T_r$ ranging from 0.22 to 1.6 with the average of 0.65. Accordingly, the observation error is larger than that of CTRL

everywhere, and this resulted in the reduction of the reconstruction skill. On the other hand, the better skill in T2-Assim should be owing to the difference in Kalman gain. The

Kalman gain determines analysis increments by spreading the information in observations through the covariance between the prior and the prior-estimated observations. We found that the correlations between the prior (temperature) and the prior-estimated observation (temperature and $\delta^{18}O$ for T2-Assim and CTRL, respectively) were consistently high in

T2-Assim than in CTRL (not shown) as Dee et al. (2016) showed. Thus, the information in the observations were more effectively spread to the analysis in T2-Assim, and this resulted in the improved skill. Note that the screening process hardly hampered the reconstruction skill, because even if the reconstructed temperature was fully used (i.e. not screened), the skills were almost the same as T2-Assim.

Conducting similar experiments, Dee et al. (2016) also concluded that the reconstruction skills were almost the same among proxy DA and reconstructed DA if the relation between the reconstructed variable and the proxy is linear. As isotope-enabled

GCMs (Schmidt et al. 2007; LeGrande and Schmidt. 2009) and observations and models for tree-rings width (D'Arrigo et al. 2008; Evans et al. 2014; Dee et al., 2016) have demonstrated non-stationarity and non-linearity, however, the relations between the proxies and climate. are non-linear and non-stationary as well. Thus, we cannot it is difficult to expect that a high SNRthe skill of reconstructed DA will be maintained over time. However, stationarity and linearity do not the same as that of proxy DA if we have to be considered if the well-defined forward proxy model is well-defined models (Hughes and Ammann, 2009). ThereforeAlthough 
[revised manuscript text omitted]

h̶b, d) precipitation, for (a̶ ̶a̶n̶d̶ ̶e̶, b) CTRL and (b̶-, d̶ ̶a̶n̶d̶ ̶f̶ ̶h̶) T2-Assim. The green dot represents the location of the proxy sampling site. The hatched area means that the correlation is not statistically significant (*p* > 0.05).

[Figure]

[Figure]

**Figure 8**

Signal to noise ratio (SNR) of the reconstructed temperature from the observation used in CTRL.

[Figure]

**Figure 9**

Temporal correlations in North America between the analysis and the truth for (a–d) temperature, and (e–h) precipitation, for experiments using different proxy networks. The green dot represents the location of the proxy sampling site. The hatched area indicates where the correlation is not statistically significant ($p > 0.05$).